# Current Practices and Existing Gaps of Continuing Medical Education among Resident Physicians in Abha City, Saudi Arabia

**DOI:** 10.3390/ijerph17228483

**Published:** 2020-11-16

**Authors:** Safar Abadi Alsaleem, Najwa Mohammed Almoalwi, Aesha Farheen Siddiqui, Mohammed Abadi Alsaleem, Awad S. Alsamghan, Nabil J. Awadalla, Ahmed A. Mahfouz

**Affiliations:** 1Department of Family and Community Medicine, College of Medicine, King Khalid University, Abha 61421, Saudi Arabia; asalslim@kku.edu.sa (S.A.A.); afarheen@kku.edu.sa (A.F.S.); mabade@kku.edu.sa (M.A.A.); asoman@kku.edu.sa (A.S.A.); njgirgis@kku.edu.sa (N.J.A.); 2Family Medicine Department, Aseer General Directorate of Health, Abha 62523, Saudi Arabia; Dr.najwa15@gmail.com; 3Department of Community Medicine, College of Medicine, Mansoura University, Mansoura 35516, Egypt; 4Department of Epidemiology, High Institute of Public Health, Alexandria University, Alexandria 21511, Egypt

**Keywords:** continuing medical education, resident physicians, Abha, Saudi Arabia

## Abstract

Background: Continuing medical education (CME) is an everlasting process throughout the physician’s working life. It helps to deliver better services for the patients. Objectives: To explore CME among resident physicians in Abha City; their current practices, their opinions, and barriers faced. Methods: A cross-sectional study was conducted among resident physicians at the Ministry of Health hospitals in Abha City using a validated self-administered questionnaire. It included personal characteristics, current CME practices, satisfaction with CME, and barriers to attendance. Results: The present study included 300 residents from 15 training specialties. Their reported CME activities during the previous year were lectures and seminars (79.7%) followed by conferences (43.7%), case presentations (39.7%), workshops (34.0%), group discussion (29/7%), and journal clubs (27.3%). Astonishingly enough, very few (8%) attended online electronic CME activities. There were significant differences in CME satisfaction scores by different training specialties. Regarding residents’ perceptions of the effectiveness of different CME activities (conferences/symposia, workshops/courses, and interdepartmental activities) the results showed that workshops and courses were significantly the most effective method compared to the other two methods in retention of knowledge, improving attitudes, improving clinical skills, improving managerial skills, and in improving practice behaviors. Barriers reported were being busy, lack of interest, high cost, and lack of suitable providers. Conclusion: Based on the findings of this study, it is recommended that online learning be promoted as a CME format for trainees. There should be support of residents and clinicians through the provision of protected time for their CME activities outside their daily clinical commitments.

## 1. Introduction

Medical professionals continuing their training throughout their careers is termed “continuing medical education” (CME) [1]. Formal CME includes organized activities such as conferences, workshops, symposia, courses, and educational meetings [2]. Personal efforts such as frequent/habitual reading and inquiries that help a person to remain up to date in his/her professional development are also included [3,4]. Currently, CME is shifting from a traditional passive model to a competency-based, self-directed learning model [5]. It is an everlasting process throughout the physician’s working life. CME helps not only to improve knowledge and skills but also to build up improved relationships to deliver better services for patients [6].

Without active learning, physicians are unable to remain competent for more than a few years after graduation [7]. For physicians to stay fit to practice, it is thus a core professional responsibility, if not an ethical and moral obligation, to continue to learn throughout their career regardless of discipline, specialty, or type [8].

Research from Australia reported that modern, technology-dependent forms of CME were less common but were believed by respondents to be the best mode of delivery [9]. Another British study reported that younger and more recently qualified doctors choose online CME, and their choice appears to be unrelated to workload and personal pressures [10]. Studies across U.K., Australia and South Africa found that traditional CME activities (lectures, conferences, and journal reading) remain the most popular forms of CME used. All doctors found lack of time as a barrier to CME activities [9,10,11].

There are many issues facing health-related human resource development in the Kingdom of Saudi Arabia, and CME is not spared from these challenges. CME involves many actors such as providers, sponsors, participants, and regulators. Several challenges have been identified including a lack of transparency in the CME budget, which leads to a too close relationship between the pharmaceutical industry and physicians, as well as issues of coordination between providers, adequate assessment of needs and resources, effectiveness, and quality control [12].

In Saudi Arabia, the Saudi Commission for Health Specialties (SCFHS) is running different postgraduate programs for training of graduates in different health specialties. By the end of the training and passing the assessment modalities, the candidate will get the Saudi Board Certificate, which is the highest professional certificate in general specialties awarded by the Commission. Each program will run for five years. Similarly, the SCFHS is running CME, which is defined as educational activities that aim to keep health practitioners abreast with the latest developments in their specialties and related fields.

In Saudi Arabia, the regulatory body that approves and accredits all CMEs, SCFHS, has ruled that all healthcare practitioners must acquire a set number of CME hours (minimum 30 h) annually for successful licensing to practice as physicians in the Kingdom. Resident physicians in different SCFHS postgraduate training programs are exempted from CME activities for finalizing their training in the due programs. Nevertheless, they need to attended the minimal CME hours to be licensed for their practice. Saudi candidates getting their postgraduate training outside the Kingdom are also required to have CME hours before being licensed to practice in Saudi Arabia. CME activities are given to practicing physicians (residents, specialists, and consultants) [3].

Essentially, this regulation presumes that attending CME events secures the improvement of attendees’ knowledge and skills [3]. However, this regulatory approach is not sufficient to ensure effective learning. There is a need to give proper attention to the principles of adult learning to enable learners to be active participants in developing training programs. It is also necessary to fashion the learning activities according to the learners’ goals [13]. In Saudi Arabia, there have been no substantial reports on such aspects of CME [3].

Previously, studies have focused on quality issues in CME rather than on the needs assessment of physicians who are the major stakeholders [14,15]. A key study has described that the development, implementation, and evaluation of an educational program in family medicine for general practitioners in Saudi Arabia showed very promising results with significant improvement of knowledge [16]. However, there is dearth of information on the CME needs of the participants. The present study aimed to explore CME among resident physicians in Abha City: their current practices, their opinions, and barriers faced.

## 2. Materials and Methods

### 2.1. Study Design and Setting

A cross-sectional study was conducted among male and female residents at the Ministry of Health hospitals in Abha City (Aseer Central Hospital and Abha Maternity and Children’s Hospital).

### 2.2. Target Population and Sampling

With an anticipated proportion of CME training among residents of 50% [3] and an absolute precision of 6% at 95% confidence, the minimal sample size needed for the survey was calculated to be 267 residents [17]. To account for potential nonresponse, an overall sample size of 300 residents was initially planned for the study. A list of residents was obtained that yielded 400 currently enrolled residents from all departments. All residents were invited to participate in the study. A total of 313 residents returned filled questionnaires. Thirteen incompletely filled questionnaires were discarded, leaving a final sample of 300 residents (response rate of 75%).

### 2.3. Survey Tool

This study used a validated self-administered questionnaire that was published before in similar cross-sectional studies in Saudi Arabia [3,18]. This tool was used in the present study to facilitate comparison with similar local studies in Saudi Arabia (sharing similar cultural and educational background). The questionnaire included personal characteristics such as age, gender, nationality, marital status, specialty, position, qualification, and years of experience. The aspects of CME that were included were current participation in CME activities, frequency of CME activities, type of activity (conference, seminar, self-reading), reasons for attending CME, satisfaction with current CME practices, preferred methods of instruction in different CME activities, and barriers to attending CME. Preferences for future CME activities (time of activity, duration, evaluation methods) were also obtained.

Each satisfaction item was given a score between 0 and 4, where strongly disagree = 0 and strongly agree = 4. The total satisfaction score was obtained by adding up the responses. Cronbach’s alpha for the eight satisfaction questions in the present study amounted to 0.85, indicating very good reliability and internal consistency.

### 2.4. Data Analysis

Data were analyzed using the SPSS software package (IBM Corp. Released 2013. IBM SPSS Statistics for Windows, Version 22.0. IBM Corp., Armonk, NY, USA). Categorical data were summarized using frequencies and percentage, while continuous data were summarized using median, range, arithmetic mean, and standard deviation (SD). Chi-square, Student “t”, and ANOVA were used as tests of significance at a 5% level of significance.

### 2.5. Ethical Approval and Consent to Participate

Written informed consent was obtained from all residents. They were given assurance of confidentiality. This study was approved by the King Khalid University Research Ethical Committee (REC: 2018/05/72). All necessary official approvals to conduct this study were obtained.

## 3. Results

The present study included a total of 300 residents. They were 176 (58.7%) males and 124 (41.3%) females. Most of them were Saudis (*n* = 289, 96.3%). Half of all residents were married (151, 50.3%). By specialty, 24.3% (73) were residents in family medicine, 21.3% (64) in internal medicine, and 12.7% (38) in pediatrics. The rest were from other specialties, namely obstetrics and gynecology (17), ear nose throat (15), dermatology (14), orthopedics (14), preventive medicine (14), general surgery (12), ophthalmology (11), radiology (10), psychiatry (6), emergency medicine (4), restorative dentistry (4), and urology (4). By the level of residency, there were 89 residents (29.7%) in their first year (R1), 75 in R2, 74 in R3, 44 in R4, and 7 in R5.

Regarding the CME activities in the previous training year, the reported activities were lectures and seminars (79.7%) followed by conferences (43.7%), case presentations (39.7%), workshops (34.0%), group discussion (29/7%), and journal clubs (27.3%). Astonishingly enough, very few (8%) attended online electronic CME activities.

Table 1 shows the average total satisfaction scores for the study sample. There were no significant differences regarding gender, marital status, nationality, and training level of residency (*p* > 0.05). On the other hand, there were significant differences in satisfaction scores by different training specialties (*p* = 0.04). The highest scores were among restorative dentistry, emergency medicine, and internal medicine. The lowest satisfaction scores were among psychiatry, general surgery, and ophthalmology.

Figure 1 describes the residents’ beliefs about CME as assessed by their level of agreement with various statements. To the statement, “I believe that my CME needs are currently satisfied,” 42.3% of the residents agreed. To the statement, “I believe that my medical school education encouraged me to be an independent self-learner,” 49.4% agreed. The majority believed that CME should be organized on a national level (68.7%), CME kept them up to date (65.0%), CME improved their practice (68.7%), CME affected their professional confidence (65.0%), CME offered new learning opportunities (65.3%), and CME provided sufficient scope for questions and discussion (57.6%).

Figure 2 describes the residents’ perceptions of the most effective method of different CME activities (conferences/symposia, workshops/courses, and interdepartmental activities). It shows that workshops and courses (combined) were significantly the most effective method compared to the other two methods in retention of knowledge, improving attitudes, improving clinical skills, improving managerial skills, and in improving practice behaviors. On the other hand, interdepartmental activities were significantly the most effective method compared to the other two methods in improving academic skills and improving departmental image.

Table 2 shows participants’ preferred methods of instruction in the CME activities. The highest frequency of residents preferred that lectures should take place mainly in conference/symposium CME activities (37%). For demonstration, hands-on practice, small group seminars, live case presentation, and simulations, residents preferred that they should take place mainly in workshops (39.7%, 54%, 36%, 29.7%, and 40.7%, respectively). For distance learning and electronic meeting, the highest frequency of residents preferred them in conferences/symposium (39.7% and 44.7%, respectively). However, these figures were low, particularly in other methods of CME activities including interdepartmental activities (9% and 10.3%, respectively) and workshops (23.3% and 17.7%, respectively).

Results of the present study regarding self-reading as a method of CME activities, showed that the most prevalent method was reading medical books (230, 76.7%), online websites (196, 65.3%), social media (91, 30.3%), and medical journals (63, 21%). When asked how often they read, the highest frequency of residents (128, 42.7%) reported weekly self-reading. The reasons for using self-reading as a CME method were ease of time management (208, 69.3%), ease of place (104, 34.7%), and low price (54, 18%). Barriers to self-reading were reported as being busy (212, 70.7%), lack of interest (31, 10.3%), lack of provided materials (26, 8.7%), and high cost (26, 8.7%).

Results concerning attending conferences and symposium CME activities, reported that the highest frequency was weekly attendance (130, 43.3%). As for the reason for choosing these methods, the highest frequent reason was suitable place and time (44.7% and 41.7%, respectively). Barriers reported were being busy (155, 51.7%) and lack of interest and lack of suitable providers (50, 16.7%, each).

Results of attending workshops and courses as CME activities revealed that the highest frequency of attendance was infrequently (134, 44.7%) followed by monthly (81, 27%). As for the reason for choosing these CME methods, suitable time and suitable place were the most frequently mentioned reasons (40.7% and 37.3%, respectively). Barriers for attendance were being busy (124, 41.3%), lack of availability (101, 33.7%), and high cost (46, 15.3%).

As for the preferred duration of CME activities, the majority of participating residents (153, 51%) selected one to two days, followed by three to seven days (137, 45.7%). The majority (194, 64.7%) preferred that the CME activity should take place on a working day. Mornings periods were preferred by most participants (216, 72%). Regarding methods of CME activity evaluation, the highest frequency of residents (135, 45%) preferred a questionnaire.

To summarize the results, resident physicians mostly attended conferences, lectures, and seminars and to a lesser extent workshops and group discussions for their CME needs. Electronic CMEs were used by a minority of them. They reported workshops and courses to be most effective in retention of knowledge, improving attitudes, improving clinical skills, improving managerial skills, and in improving practice behaviors. The reported barriers to CME were being busy, lack of provision, lack of suitability, and high cost.

## 4. Discussion

Continuing medical education (CME) constitutes a wide range of educational activities that aim for the maintenance, development, and improvement of the knowledge, skills, and professional performance that enable practicing clinicians to provide safe and effective clinical services. Lifelong learning remains an indispensable element in enhancing clinical knowledge and professional behavior among practicing clinicians [19], as clinical experience over the years does not necessarily yield higher levels of skills, professional behavior, or knowledge [20,21]. The present study explored various aspects of CME activities utilized by resident physicians in Abha City in Saudi Arabia, as well as their opinions of the effectiveness of CME activities and the barriers that hinder accessing them.

This study found that the most popular CME activity among the participating residents was lectures and seminars. Lectures are not just popular in our sample of trainees, but also remain the main source for CME. Around the world, studies exploring physicians’ CME preferences agreed with our study findings [6,22]. An exploratory study from Saudi Arabia confirmed that the majority of healthcare professionals preferred symposia and short courses [3]. Indeed, lectures are famous as one of the most common forms of dispensing knowledge among clinicians and have been found to have a positive impact on knowledge and skills, though they have very little performance-improving effect [6]. The least popular CME activity was electronic activities. These are non-contact CME activities, which could be one of the reasons for their unpopularity among trainees [23]. Research has found that the more interaction there is between the adult learner and the educator, the greater the satisfaction with the learning method [22]. Electronic CME activities have certain sophisticated requirements, such as access to smart devices and software and the ability to afford extensive data connectivity costs [24]. Electronic CME activities have many advantages, such as flexibility and individualized learning format [23,25], and trainees should be encouraged to make effective use of them. Notably, some studies did not find substantial differences between didactic lectures and online teaching methods in improving diagnostic skills among physicians [26,27].

In the current study, the resident physicians did not show much inclination toward using live casts or electronic conferencing, particularly for workshops, courses, or interdepartmental activities. This is an interesting finding, considering that the internet penetration rate in KSA, at 93.3%, is among the highest globally [28]. This highlights under-usage of an effective resource-friendly learning method, which has been shown to improve the knowledge, attitude, and practice of physicians [29]. This trend has been reported earlier by students from a university in Najran, Saudi Arabia, who reported that 97% of them have computer and internet access, yet 73% opted for conventional learning methods [30].

One of the main findings of the present study is that the main barrier against attending all forms of CME was being busy, although it was higher for self-reading than for contact-based CME such as attending lectures and seminar courses. Our findings are a stark reminder of the reality of how busy clinical services are nowadays. No matter how motivated a clinician is, time pressures can hamper their CME goal attainment. This finding has been confirmed across national and international studies [31], and indeed little has changed in the past two decades, as a study in the same region some 20 years ago revealed that physicians had little time allocated for CME practice [32]. Time constraints were the second most common reason for not attending CME activities in a Pakistani sample of physicians [33]. The problem of a lack of protected time, we suspect, must have increased in recent times. Contrary to our findings, a recent study from the Eastern Region of Saudi Arabia found that lack of postgraduate education and dissatisfaction with CME activities were the main reported barriers to physicians’ accessing CME activities [24]. Interestingly, that study found that high case load is an impetus for physicians to gain extra knowledge and skills and engage in CME more frequently [34].

Another reassuring finding in this paper is that almost half of the participants affirmed that conferences improved their clinical practice and academic skills. Moreover, two out of every three residents confirmed that workshops and courses improved their clinical skills. It is expected that improving online interactions could increase the effectiveness of these CME methods, which is very promising in the future in the capacity building process of the residents. This overall improvement in clinical skills as an effect of engaging in CME activities has been well established in several educational papers [35,36,37,38] and is consistent with the results of the current study. The results of our study are grounds for cautious optimism and point to improvements in physicians’ satisfaction with CMEs over that reported in the last decade [18].

This study did not establish any background factors exerting a significant impact on satisfaction with CMEs, which agrees with the results of another study [39]. It is reassuring that gender was not a determining factor in satisfaction with CME activities. Only few differences were observed between training specialties, and it was worrying that psychiatric trainees were the least satisfied among the participants. Many surveys indicate that dental health professionals are in general more satisfied with CME activities than other healthcare workers [38]. This could explain the differences in satisfaction scores between trainees in restorative dentistry and other medical trainees observed in the current study.

The current study provided some much-needed answers. While it was comprehensive in assessing the CME practices and needs of the residents, it is limited by the fact that the study concerns resident physicians of one region of Saudi Arabia. Generalization of the present study should be performed with some caution. Another limit to our study was that it involved only resident physicians at governmental care hospitals in Abha City.

## 5. Conclusions

CME should go beyond the sheer acquisition of knowledge. It should also carry out changes in the practice, attitude, and professional behavior of physicians. The CME offerings are subject to the goals of the organizing institution. Furthermore, it should meet with the needs and desires of the end user. Based on the results of the current study, it is recommended that online learning be promoted as a CME format for trainees and practicing clinicians in Saudi Arabia. There should be support of residents and clinicians through the provision of protected time for their CME activities outside their daily clinical responsibilities. Future research should focus on the effects of various forms of CME on clinical effectiveness as well as on the professionalism and communication skills of physicians.

## Figures and Tables

**Figure 1 ijerph-17-08483-f001:**
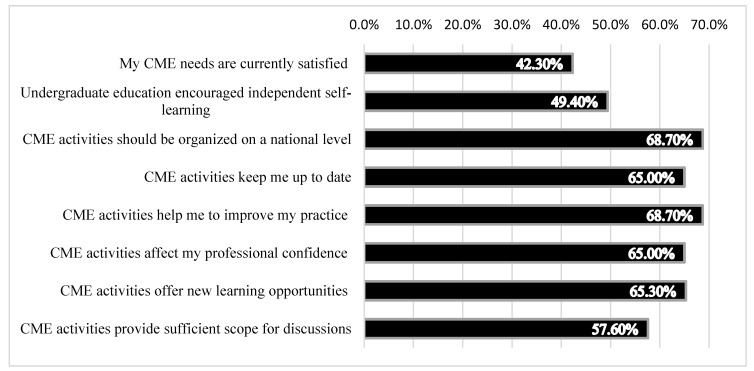
Resident’s beliefs about continuous medical education.

**Figure 2 ijerph-17-08483-f002:**
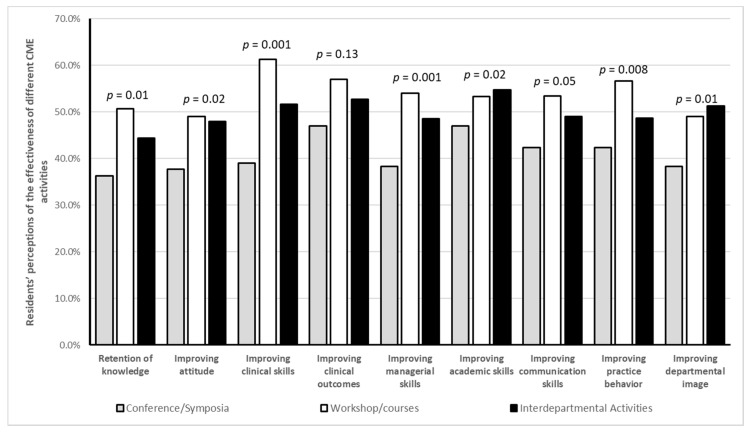
Residents’ perceptions of the most effective method among different CME activities.

**Table 1 ijerph-17-08483-t001:** Total satisfaction score with continuing medical education (CME) activities among the study residents by their characteristics (*n* = 300).

Characteristic	Total Satisfaction Score (Mean ± SD)	*p*-Value
Gender		
Males	21.01 ± 5.31	0.98
Females	21.03 ± 5.84	
Marital status		
Married	21.20 ± 5.44	0.59
Single	20.85 ± 5.65	
Nationality		
Saudi	21.02 ± 5.54	0.92
Non-Saudi	21.18 ± 5.45	
Training specialty		0.04
Family Medicine	21.88 ± 5.19	
Internal Medicine	22.41 ± 5.35	
Pediatrics	19.39 ± 6.93	
Obstetrics and Gynecology	20.76 ± 6.09	
Ear Nose and Throat	20.07 ± 4.30	
Dermatology	21.29 ± 5.12	
Orthopedics	21.00 ± 4.52	
Preventative Medicine	19.21 ± 3.33	
General Surgery	18.83 ± 6.71	
Ophthalmology	19.09 ± 6.71	
Radiology	21.00 ± 3.62	
Psychiatry	16.00 ± 5.37	
Emergency Medicine	24.50 ± 4.43	
Restorative Dentistry	26.00 ± 1.83	
Urology	20.00 ± 9.09	
Training level of residency		0.54
R1	20.27 ± 4.85	
R2	21.19 ± 6.72	
R3	21.47 ± 5.14	
R4	21.82 ± 5.14	
R5	21.29 ± 5.59	

**Table 2 ijerph-17-08483-t002:** Residents’ preferred methods (*n* = 300) of instruction in the CME activities during the past one year.

Preferred Instruction Methods	CME Activities			
	Conference/Symposium No. (%)	Workshop No. (%)	Courses No. (%)	Interdepartmental Activities No. (%)
Lecturing	77 (25.7)	60 (20.0)	52 (17.3)	111 (37.0)
Demonstration	24 (8.0)	60 (20.0)	119 (39.7)	97 (32.3)
Hands-on practice	39 (13.0)	49 (16.3)	162 (54.0)	50 (16.7)
Small group seminar	72 (24.0)	68 (22.6)	108 (36.0)	52 (17.4)
Live case presentation	74 (24.7)	50 (16.7)	89 (29.7)	88 (29.3)
Simulations	66 (22.0)	55 (18.3)	123 (40.7)	57 (19.0)
Distant learning	27 (9.0)	84 (28.0)	70 (23.3)	119 (39.7)
Electronic conferencing	31 (10.3)	82 (27.3)	53 (17.7)	134 (44.7)

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
