# Peer review of "Current Practices and Existing Gaps of Continuing Medical Education among Resident Physicians in Abha City, Saudi Arabia"

_ijerph, 2020, doi:10.3390/ijerph17228483_

Round 1

Reviewer 1 Report

Review of “Current Practices and Existing Gaps of Continuing Medical Education Among Resident Physicians in Abha City, Saudi Arabia”

Congratulations to authors for conducting this research.

Many important points for this field of knowledge are very relevant for scientific community.

However, some points need to be considered:

  1. How did you find these frequency outputs in lines 188 and 193?
  2. Clearly, considering table 4, Conference/Symposium and workshops are preferred modalities towards instruction methods. Why the authors considered courses?
  3. Concerning training specialties table 1, would be possible to infer that the article results in the conclusion, if not incorrect (workshops and courses as preferred modalities), could be observed for the entire types of specialties? As authors noticed, dentistry health have its specific interests and the same can happen to each one of the specialties.
  4. How did this data was filtered in terms of the specialties and general conclusion “workshops and courses are preferred by the residents”?
  5. In line 212, this sentence “This study found that the most popular CME activity among the participating residents was lectures and seminars.”, this is a conclusion of the study or a reference? Because the study results didn’t present it clearly.
  6. Here at this sentence at line 228, “In the current study, the resident physicians did not show much inclination towards using live 229 casts, or electronic conferencing, particularly for workshops, courses or inter departmental activities.”, this could be very different if specialties would take in place one by one isn’t? If yes, the results are much more conclusions from a biased sample than majority of patterns forming in the sample isn’t? Please address this question.
  7. About the “data availability”, is this data stored online or in a medical facility?
  8. Looking at table 4 results, it seen that conference/symposium have high interest and could be very well improved with online interactions. In this sense, not only workshops and course, isn’t?

Author Response

Kindly find attached the responses

Reviewer 2 Report

This is a very interesting study.  Thank you for sharing.  I would suggest considering presenting your results in a format other than just tables.  Something that might provide a better visual understanding of the material.  Tables are difficult to interpret without spending time.  It would be helpful to use graphs or figures to get the results across quickly.  

Author Response

Kindly find attached the responses

Reviewer 3 Report

1.
The literature review is correct. Research gaps were also rightly identified in other studies conducted
2.
There is no emphasis here on why such research methodology was used in this research tool. What was the guiding principle behind this decision?
This should be completed.
3.
The research results are presented in a very large format, but at the end there should be a short synthesis. This will close this part of the article and open the discussion
4.
This part contains the content that is missing from the previous one. On the other hand, the Discussion should be the last element of the article and refer primarily to possible limitations of research and looking into the future at the continuation of the issue
5.
And again, here are the content that should be in the Discussion, I propose to combine the conclusions and the discussion, of course without the content that should be transferred earlier.

In conclusion, the article is very interesting, but it showed the authors' shortcomings in the correct preparation of the presented content. Most of the amendments will reformat the content between chapters.

Author Response

Kindly find attached the responses

Round 2

Reviewer 1 Report

Dear authors, thank you for your review.

The new images seemed to substitute some tables.

They are good enough.

The few amendments were enough to clear the doubts about the research methodology.

Reviewer 3 Report

I accept the explanations of the authors, although the final part of the text should still be reorganized and changed
